# Is Evar Feasible in Challenging Aortic Neck Anatomies? A Technical Review and Ethical Discussion

**DOI:** 10.3390/jcm11154460

**Published:** 2022-07-30

**Authors:** Pasqualino Sirignano, Silvia Ceruti, Francesco Aloisi, Ascanio Sirignano, Mario Picozzi, Maurizio Taurino

**Affiliations:** 1Vascular and Endovascular Surgery Unit, Sant’Andrea Hospital of Rome, Department of Surgery “Paride Stefanini”, “Sapienza” University of Rome, 00189 Rome, Italy; 2Center for Clinical Ethics, Department of Biotechnologies and Life Sciences, University of Insubria, 21100 Varese, Italy; cerutisilvia@gmail.com (S.C.); mario.picozzi@uninsubria.it (M.P.); 3Vascular and Endovascular Surgery Unit, Sant’Andrea Hospital of Rome, Department of Molecular and Clinical Medicine, “Sapienza” University of Rome, 00189 Rome, Italy; fra.aloisi@gmail.com (F.A.); maurizio.taurino@uniroma1.it (M.T.); 4School of Civil Law, University of Camerino, 62032 Camerino, Italy; ascanio.sirignano@unicam.it

**Keywords:** abdominal aortic aneurysm, EVAR, hostile anatomy, EVAR outcomes, IFU, instruction for use, ethical considerations, informed consent, shared decision-making

## Abstract

Background: Endovascular aneurysm repair (EVAR) has become an accepted alternative to open repair (OR) for the treatment of abdominal aortic aneurysm (AAA) despite “hostile” anatomies that may reduce its effectiveness. Guidelines suggest refraining from EVAR in such circumstances, but in clinical practice, up to 44% of EVAR procedures are performed using stent grafts outside their instruction for use (IFU), with acceptable outcomes. Starting from this “inconsistency” between clinical practice and guidelines, the aim of this contribution is to report the technical results of the use of EVAR in challenging anatomies as well as the ethical aspects to identify the criteria by which the “best interest” of the patient can be set. Materials and Methods: A literature review on currently available evidence on standard EVAR using commercially available endografts in patients with hostile aortic neck anatomies was conducted. Medline using the PubMed interface and The Cochrane Library databases were searched from 1 January 2000 to 6 May 2021, considering the following outcomes: technical success; need for additional procedures; conversion to OR; reintervention; migration; the presence of type I endoleaks; AAA-related mortality rate. Results: A total of 52 publications were selected by the investigators for a detailed review. All studies were either prospective or retrospective observational studies reporting the immediate, 30-day, and/or follow-up outcomes of standard EVAR procedures in patients with challenging neck anatomies. No randomized trials were identified. Fourteen different endo-grafts systems were used in the selected studies. A total of 45 studies reported a technical success rate ranging from 93 to 100%, and 42 the need for additional procedures (mean value of 9.04%). Results at 30 days: the incidence rate of type Ia endoleak was reported by 37 studies with a mean value of 2.65%; 31 studies reported a null migration rate and 32 a null conversion rate to OR; in 31 of the 35 studies that reported AAA-related mortality, the incidence was null. Mid-term follow-up: the incidence rate of type Ia endoleak was reported by 48 studies with a mean value of 6.65%; 30 studies reported a null migration rate, 33 a null conversion rate to OR, and 28 of the 45 studies reported that the AAA-related mortality incidence was null. Conclusions: Based on the present analysis, EVAR appears to be a safe and effective procedure—and therefore recommendable—even in the presence of hostile anatomies, in patients deemed unfit for OR. However, in order to identify and pursue the patient’s best interest, particular attention must be paid to the management of the patient’s informed consent process, which—in addition to being an essential ethical-legal requirement to legitimize the medical act—ensures that clinical data can be integrated with the patient’s personal preferences and background, beyond the therapeutic potential of the proposed procedures and what is generically stated in the guidelines.

## 1. Introduction

Endovascular aneurysm repair (EVAR) has become an accepted alternative to open repair (OR) for the treatment of abdominal aortic aneurysm (AAA), with >75% of AAA repairs performed endovascularly [1]. Undoubtedly, EVAR is associated with lower 30-day mortality and morbidity, faster discharge, and fewer complications than OR [2], but some concerns remain about durability, need for long-term follow-up, and reinterventions [1,2,3].

Based on preclinical engineering assessments and clinical study results, particular anatomical characteristics, specifically aortic neck diameter, length, angle, and shape, are recommended to guide patient selection for EVAR [4]. Indeed, unfavorable anatomy, documented in 40–60% of treated AAAs, seems to be the factor most related to negative outcomes [5,6,7]. Despite this, in “real-world” clinical practice, up to 44% of EVAR cases are performed using stent grafts outside their instruction for use (IFU) due to the presence of a ”hostile” aortic neck anatomy, with acceptable short- and mid-term outcomes [8,9,10]. In order to overcome this problem, several different technical solutions, such as parallel grafts and fenestrated or branched devices, have been developed as an alternative to standard EVAR grafts in patients with a “challenging neck”. However, all those solutions present a relevant risk of reintervention due to gutter-related endoleaks and target vessel instability, and higher cost compared to standard EVAR; lastly, custom-made devices are not suitable for urgent/emergent cases [7].

Nevertheless, because EVAR durability is related to the maintenance of a seal between the stent graft and the aortic neck, some authors suggested that challenging the neck could affect long-term outcomes, increasing the risk of type Ia endoleak, reintervention, and aneurysm-related mortality rates [11,12,13,14,15]. As a result of these long-term results, current guidelines suggest limiting or even refraining from adopting EVAR in patients with challenging aortic necks [16,17,18,19].

Moving from this inconsistency between “real world practice” and “best practice” suggested by current guidelines [20], the purpose of this contribution is to report technical and clinical results of the use of EVARs in challenging anatomies and to analyze and discuss the ethical implications of implementing these procedures, to identify criteria by which the patient’s “best interest” should be defined in these specific circumstances.

## 2. Material and Methods

### 2.1. Search Strategy

A literature review on currently available evidence on standard EVAR using commercially available endografts in patients with hostile aortic neck anatomies was conducted by three investigators (F.A., S.C., P.S.), and an eligibility assessment of studies for inclusion in this review was performed in a non-blinded standardized manner by the same investigators. Disagreements were resolved by discussion and consensus.

Medline using the PubMed interface and Cochrane Library databases were searched from 1 January 2000 to 6 May 2021 using the search strategies {*“EVAR”[All Fields] AND “AAA”[All Fields] AND (“neck”[MeSH Terms] OR “neck”[All Fields]) AND (“hostile”[All Fields] OR “hostiles”[All Fields] OR “hostility”[MeSH Terms] OR “hostility”[All Fields] OR “hostilities”[All Fields] OR (“challenge”[All Fields] OR “challenged”[All Fields] OR “challenges”[All Fields] OR “challenging”[All Fields]) OR (“short”[All Fields] OR “shorts”[All Fields]) OR (“angulate”[All Fields] OR “angulated”[All Fields] OR “angulates”[All Fields] OR “angulating”[All Fields] OR “angulation”[All Fields] OR “angulational”[All Fields] OR “angulations”[All Fields]) OR (“conic”[All Fields] OR “conical”[All Fields] OR “conically”[All Fields] OR “conics”[All Fields]) OR “wide”[All Fields] OR (“large”[All Fields] OR “largely”[All Fields] OR “larges”[All Fields])*}.

A further search was undertaken including a manual screen of the reference lists of selected articles identified through the electronic search; only English papers were considered for the present review.

### 2.2. Study Selection: Study Design and Data Extraction

Studies included in the analysis met the following criteria: (a) randomized controlled studies, non-randomized studies, and observational studies; (b) published as original research from 2000 to 2021; (c) clearly reported the results of the patient treated in the presence of “hostile” neck anatomy; (d) described the results of standard EVAR procedures in the immediate postoperative period, and/or at 30 days, and/or at mid- or long-term follow-up.

According to ESVS and SVS guidelines for EVAR and the majority of manufacturers’ IFU currently available, the proximal aortic neck was defined as “hostile” in presence of one of more of the following criteria: *(1) short*, proximal suitable landing zone length <15 mm (or <10 mm); *(2) wide*, aortic neck diameter greater than >26 mm (or >28, >30 mm, >31 mm, >32 mm); *(3) noncylindrical shape*, tapered, reverse tapered, hourglass, barrel, bulged, and conical neck (neck dilated over 2 mm within 10 mm below the most caudal renal artery); *(4) angulated*, >60° between the long axis of the aneurysm sac and juxta-renal aorta (infrarenal angulation); *(5) thrombosed*, the widest part of thrombus (≥2 mm thick) covering at least 50% of the circumference of the proximal neck; *(6) calcified*, calcification accounting for more than or equal to 50% of the proximal neck [16,17,21].

Papers regarding endovascular abdominal sealing (EVAS) procedures were excluded due to the completely peculiar sealing mechanism of the graft and the unsatisfactory results leading to market withdrawal [22]. Case reports, letters to the editor, and conference abstracts papers were all excluded from the present analysis.

The methodological quality of studies was evaluated independently by the same three Investigators with the Newcastle–Ottawa scale, which was used to assess the quality of non-randomized studies [23].

Data were extracted into a standard Microsoft Excel file by three independent authors (F.A., S.C., P.S.) as follows: first author, year of publication, country, study design, data source/institution, duration of the study, age, number of patients, criteria, and outcomes.

### 2.3. Outcomes

Outcomes considered for the present analysis were EVAR technical success, need for unplanned adjunctive procedures, conversion to open repair reintervention, stent graft migration, type I endoleak occurrence at 30 days, and at mid/long-term follow-up were recorded, as well as AAA-related mortality at same time intervals.

### 2.4. Statistical Analysis

The data are reported as mean and standard deviation (SD) or as absolute frequencies and percentages (%). All analyses were calculated using SPSS version 25 (IBM Corp, Armonk, NY, USA).

## 3. Results

Cumulatively, a total of 172 publications were identified and their titles and abstracts were reviewed. Of these, 52 publications were selected by the investigators for a detailed review before the face-to-face panel meeting and selected for the current report (Figure 1).

### 3.1. Study Characteristics

All studies were either prospective or retrospective observational studies reporting the immediate, 30-day, and/or follow-up outcomes of standard EVAR procedures in patients with challenging neck anatomies. No randomized trials were identified. The study population ranged from 12 to 1189 patients, and the period during which selected studies were published extended from 2003 to 2021 (Table 1).

Fourteen different endograft systems were used in selected studies (Table 1): Ancure (Guidant Cardiac and Vascular Division, Menlo Park, CA, USA), AneuRx, Talent, and Endurant (Medtronic, Santa Rosa, CA, USA), Zenith (Cook, Bloomington, IN, USA), Endofit (EndoMed, Phoenix, AZ, USA), Excluder (W.L. Gore and Ass, Flagstaff, AZ, USA), Fortron (Cordis, Hialeah, FL, USA), Jotec Tube (Jotec GmbH, Hechingen, Germany), Lifepath (Edwards Life Sciences, Irvine, CA, USA), Powerlink, AFX, Ovation (Endologix, Irvine, CA, USA), and Vanguard (Boston Scientific, Marlborough, MA, USA).

Almost all the patients included in this review underwent a surveillance imaging protocol consisting of computed tomographic angiography (CTA). The great majority of the selected studies referred to guidelines suggested criteria to define a challenging neck anatomy [16,17,18,19].

### 3.2. Technical Success and Adjunctive Procedures

Forty-five studies reported the technical success rate, defined as the successful introduction and deployment of the stent grafts in the absence of surgical conversion, type I or III, or renal artery coverage. Technical success rates ranged from 89.2% to 100%; in two studies failure was due to unintentional renal artery coverage and no- to high-flow endoleak [53,55].

Forty-two studies reported the need for adjunctive procedure rate during EVAR procedures to achieve proximal seal: those procedures were either represented by needs for repeated proximal fixation site ballooning, aortic cuff or endoanchor implantation, and rescue chimney procedures. The necessity for adjunctive procedures ranged between 0 and 51%, with a mean value of 9.04%.

### 3.3. Thirty-Day Results

Regarding 30-day results, the type Ia endoleak incidence rate was reported by thirty-seven studies and ranged between 0 and 27.3% (mean value 2.65%); 23 studies reported an incidence <2%, and all but one an occurrence lower than 9% [10]. No endograft migration was reported, according to data available in 31 studies out of the 52 included (Table 2).

Reintervention due to type Ia endoleak occurred in 0–15.2% of cases (mean value 1.66%) with 30 out 37 studies recording an incidence <2%. Regarding conversion to open repair, data are reported by thirty-four studies: in 32 published experiences, no conversions were performed at a 30-day follow-up interval, while two studies [53,59] reported approximately a 2% rate (Table 2).

Lastly, data about AAA-related death rate were reported by thirty-five studies: in the great majority of them, the incidence was null, while 4 studies reported an AAA-related death rate ranging between 0.5 and 1.3%, in absence of aneurysm sac rupture after EVAR [24,32,37,44]. It is noteworthy that in the experience of Troisi et al. [32], mortality was observed only in patients initially treated for ruptured AAA (Table 2).

### 3.4. Mid-Term Follow-Up Results

All but three studies [43,60,70] reported a follow-up period longer than 30 days. Follow-up duration was evaluated by polling the data of the considered study: mean follow-up was 27.38 months (range 1–120, SD ± 23,12; Figure 2).

The type Ia endoleak incidence rate was reported by forty-eight studies and ranged between 0 and 14.8% (mean value 6.65%); half of the studies reported an incidence <2%, and seven an occurrence greater than 10% [10,25,33,40,51,58,69]. Endograft migration rate was reported with a mean value of 1.86% by 44 studies, 30 of them not reporting cases of migration. However, three studies observed an incidence >10% [25,47,69] (Table 2).

Reintervention due to type Ia endoleak occurred in 0–24.1% of cases (mean value 4.38%) with 25 studies recording an incidence <2%. Regarding conversion to open repair, data are reported by forty-one studies: in 33 published experiences, no conversions were performed at the last follow-up visits, while two studies [58,68] reported a conversion rate greater than 5% (Table 2).

Lastly, data about AAA-related death rates were reported by forty-five studies: in 28 of them, the incidence was null, while 17 studies reported an AAA-related death rate ranging between 0.6 and 11% with only Abbruzzese and coworkers reporting an incidence >10% [29] (Table 2).

## 4. Discussion

After a careful and extensive analysis of the existing scientific literature, the main result of the present contribution is that despite the above-mentioned inconsistency between “real world practice” and “best practice” suggested by current guidelines, standard EVAR in patients presenting a so-called hostile or challenging proximal aortic neck anatomy could be safely performed. Consistent with those findings, a recently published systematic review confirmed a significant reduction in perioperative mortality for EVAR compared to open repair without showing differences in AAA-related mortality at mid-term follow-up [71]. However, the risk of reinterventions still represents a major issue in those kinds of procedures and should be properly assessed in the decision-making process [71].

Of course, not all patients could be a candidate for this type of treatment, and open repair still could be considered the standard of care in patients fit for surgery [16,17,18,19]. However, from a technical point of view, standard EVAR could be safely and effectively performed (and consequently proposed) to those patients judged unfit for open surgery.

Nevertheless, a word of caution is needed regarding patients with the concurrence of multiple anatomical characteristics affecting the EVAR feasibility [7], and those presenting a wide aortic neck diameter (<30 mm) [10,51,69].

### 4.1. Ethical Considerations

The technical feasibility of standard EVAR by itself is not sufficient to exclude all ethical implications related to performing an elective procedure outside the IFU, even in fragile patients unfit for open surgery. Indeed, even if the present review shows various scientific evidence confirming that standard EVAR can be proposed even outside the IFU to patients ineligible for open surgery, identifying the most appropriate therapeutic procedure for every single patient remains a challenging issue requiring careful case-by-case evaluation for patient’s best interests.

Although this type of evaluation should always be considered an essential part of good clinical practice, both close attention to ethical requirements and acquiring a patient’s proper informed consent play a crucial role in approaching a so fragile subgroup of patients, due to the aforementioned “inconsistency” between clinical practice and guidelines [72]. In this scenario, correctly informing the patient, in addition to being an essential ethical-legal requirement to legitimize the medical act, allows integration in the decision-making process those results that are important for the patient, beyond the theoretical advantages of each proposed procedure and guideline statement [73].

As is known, the traditional “paternalistic” approach, in which the doctor was considered autonomously capable to decide for the patient’s best interest, has given way to a more holistic approach in which the patient’s autonomy and self-determination are completely integrated into the decision-making process. This need to develop more patient-centered healthcare has led to a redefinition of the concept of patient’s best interest, in which physician and patient are both equally involved as decision-makers. This shared decision-making approach requires integrating the best available medical evidence with the patient’s values, beliefs, and preferences, in an ongoing dialogic process, that promotes high-quality health care decisions from both an objective (physician), and a subjective (patient) perspective [74]. Moreover, preliminarily identifying patient priorities is essential to ensure that these aspects could be evaluated in clinical practice [75].

However, recognizing the value of patient preferences and their relevance to medical decision-making can be difficult, especially when they differ from classical clinical outcomes. In other words, faster discharge times, or the absence of postoperative discomfort could be considered extremely important by the patient (and therefore have a greater value in the decision-making process) than the crude mortality rate [76]. Notably, even studies specifically designed to address the patient’s perspective (usually based on “quality of life” as a quantitative equivalent) are essentially based on items defined by health professionals, which may not reflect the patient’s perspective. For example, these studies may not capture the patient’s “concerns about symptoms”, “the impact of possible outcomes/complications”, as well as issues related to “self-control and decision-making” [73].

Consequently, to enable patients to make decisions that are fully consistent with their individual preferences, physicians should strive to properly inform about the different treatment options, and the risks and benefits associated with each alternative [77], not forgetting the patients’ perspective [74].

All the above is necessary and, at the same time, particularly complex to obtain in fragile AAA patients unfit for open surgery and presenting with hostile aortic anatomies. Approaching such a patient, vascular surgeons are requested to discuss all treatment options (F/B-EVAR, Ch-EVAR, off-label use of EVAR, and even non-intervention), and their relative risks and benefits, personalizing information for every single patient [78,79].

Moreover, the informed consent process requires not only that the physician inform the patient, but also that the patient fully understands the information provided. Therefore, the information must be presented consistently with the understanding of each individual patient, based on her/his education level, age, and psychological and emotional status [20]. Physicians should not rush the patient’s decision: the patient’s informed consent process is, in fact, a “process” and not a simple “act”. The physician–patient relationship must be established and strengthened through a dialogue that requires commitment and time. Physicians should encourage the patient to reflect on her/his preferences, values, and goals, ask for more information, express her/his doubts, discuss with relatives, and seek a second opinion in case of uncertainty [74].

Lastly, any deficiency in the informed consent acquiring process undermines the legitimacy of the consent itself, breaks the relationship of trust between patient and physician, and potentially leads to litigations [80]. On the contrary, the more detailed the information is, the more actively the patients are involved, and the more likely they are satisfied with their decisions and their expectations are met [74,81].

Despite all these considerations, patients’ information needs are not always satisfied in everyday clinical practice: patients, especially those presenting AAA, complain about a lack of information on the treatment option and relative risks and benefits [82].

With the aim of overcoming this *vulnus* in the patient–physician relationship, several interventions were implemented to improve the quality of informed consent and to foster patient understanding of treatment options and outcomes [83,84,85,86].

Physicians should consider what patients really want to know, and what information is truly useful for them to make a decision. It is not always necessary to report everything, especially when it comes to complex clinical or statistical data. However, the physician must recognize that some patients want to be thoroughly informed about treatment—different options, and risks and benefits associated with each available option—some others prefer to receive less information, and others choose to receive no information, exercising their “right not to be informed” [72,87].

### 4.2. Study Limitations

First of all, the present study is a narrative review and not a systematic review, consequently, the statistical power of here presented data should be carefully evaluated. Moreover, not all included studies reported results on standard EVAR performed in standard anatomies, therefore, a proper comparison was not made between patients presenting with standard and hostile anatomies treated by the same operators, in the same centuries, during the same period.

### 4.3. Conclusions

In conclusion, it is not possible to establish a priori what is in the best interest of the patient; it is not possible when there is reliable scientific evidence, and it is even less so in case of inconsistency between “real world practice” and “best practice” suggested by current guidelines such as in the reported clinical scenario. In those complex cases, physician experience, available data from reviews, guidelines’ recommendations, and patients’ preferences should all be considered and carefully evaluated to reach a joint decision and to choose the right tailored approach for every single case.

## Figures and Tables

**Figure 1 jcm-11-04460-f001:**
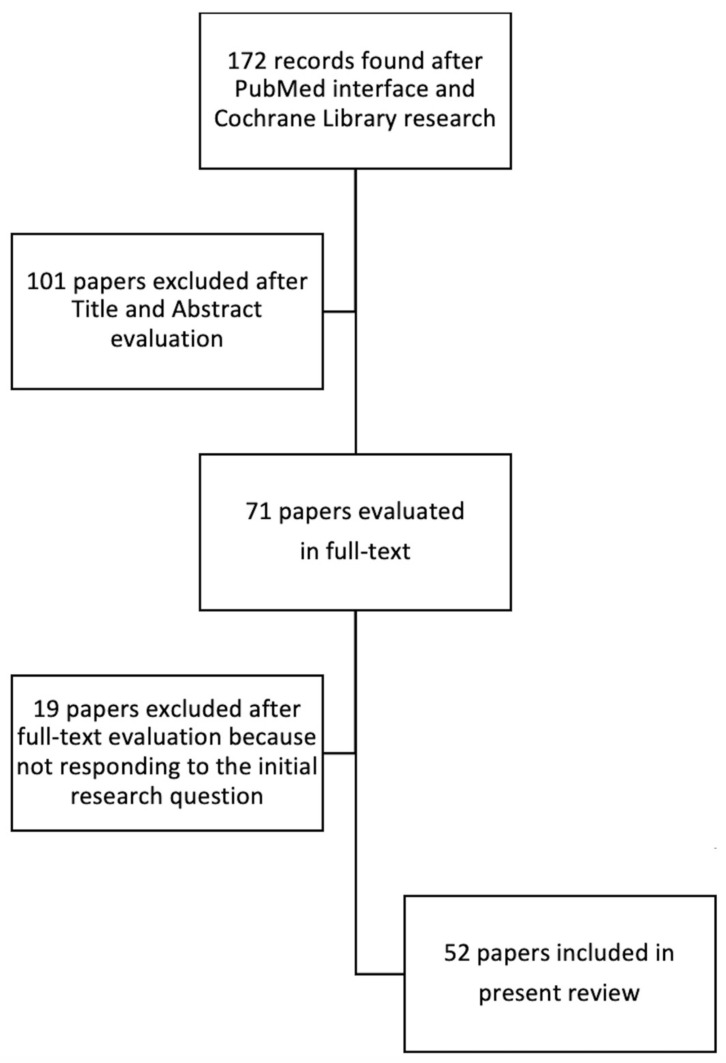
Flow chart reporting studies evaluation and selection process.

**Figure 2 jcm-11-04460-f002:**
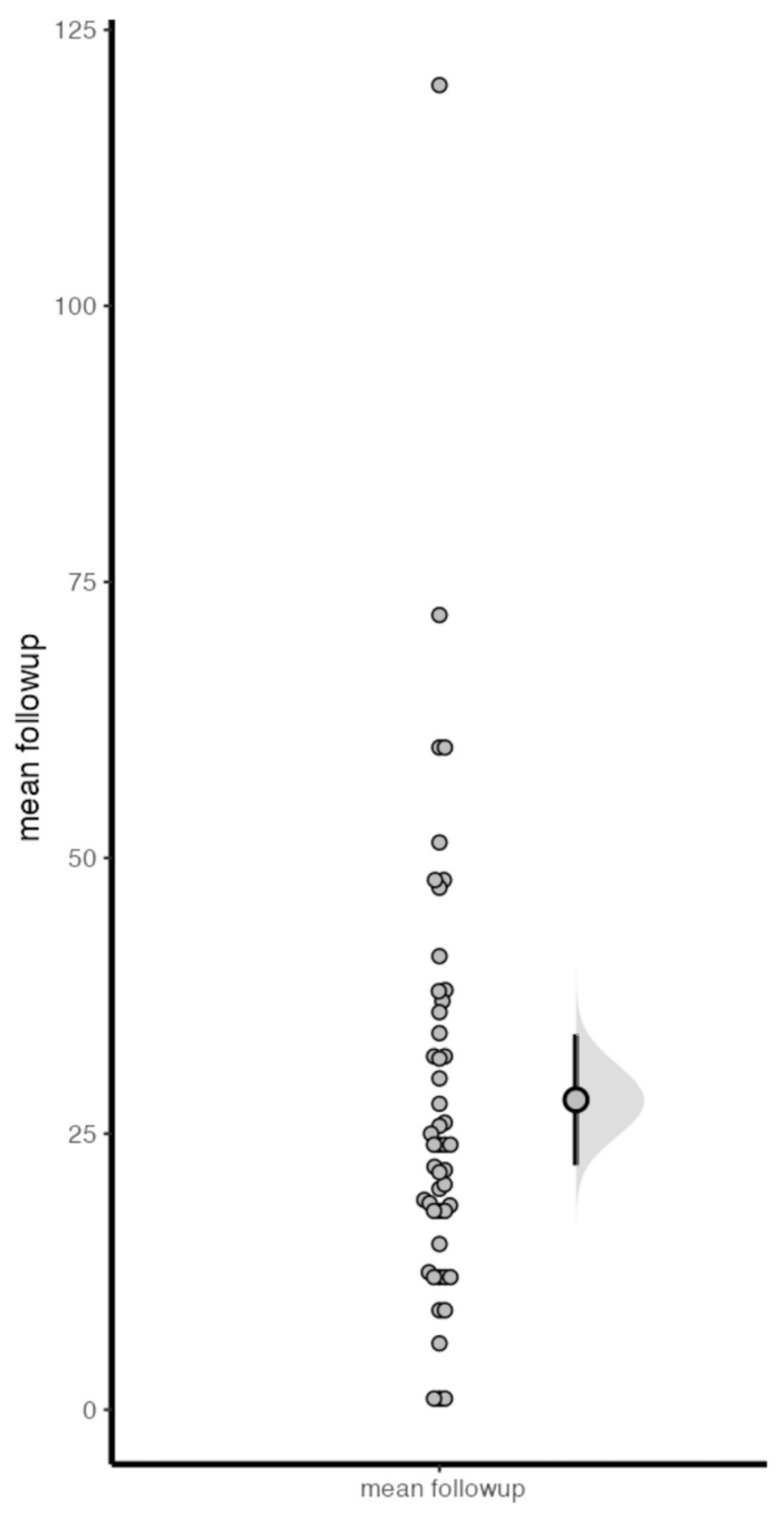
Pool of all follow-up duration data for all studies considered in the present analysis.

**Table 1 jcm-11-04460-t001:** Authors, years of publication, countries of origin, study period, number of patients, follow-up duration, type of endograft implanted, mean age, technical success, and need for adjunctive procedure rates of the 52 studies selected for the review.

First Author	Year of Publication	Country	Study Period	Number of Patients	Mean Follow-Up (Month)	Hostile Neck Anatomy Definition	Implanted Endografts	Mean Age (Years)	Technical Success (%)	Need for Adjunctive Procedures (%)
Dillavou ED [24]	2003	United States of America	1999–2002	91	18	short neck <10 mm; bulge: focal enlargement of the aneurysm neck; reverse taper: dilation >2 mm within the first 10 mm; angulated neck >60 degrees; significant neck thrombus covering > 50% of the circumference	Ancure	75.7	95.61	4.39
Fairman RM [25]	2004	United States of America	NA	153	21	short neck: <15 mm; very short neck: <10 mm, dilated neck >28 mm, angulated neck > 45 degrees; calcified, and thrombus-lined, with or without ulceration	Talent	NA	NA	NA
Choke E [26]	2006	United Kingdom	1997–2005	60	21.7	short neck:< 10 mm; wide neck: >28 mm; angulated neck: >60 degrees; significant neck thrombus covering > 50% of the circumference	AneuRX, Excluder, Zenith, Talent, Fortron, Endofit, Vanguard, Lifepath	74.4	98	18
Cox DE [27]	2006	United States of America	2000–2004	19	12	short neck: <15 mm; wide neck: >26 mm; angulated neck: >60 degrees; circumferential neck thrombus; neck bulge; reverse taper neck: dilated >2 mm within 10 mm	Zenith, Aneurx	72	100	15.78
Mc Donnell CO [28]	2006	Australia	2001–2004	46	20.2	Flared neck; Barrel neck; cone neck; irregular neck; hourglass neck	Talent, Zenith	NA	NA	NA
Abbruzzese TA [29]	2008	United States of America	1999–2005	222	29.6	any deviation from single-device IFU	Zenith, Excluder, AneuRX	NA	NA	NA
Chisci E [30]	2009	Italy, Sweden	2005–2007	74	19	neck diameter >28 mm; neck length <15 mm, neck angulation >60 degrees; reverse, tapered or bulging neck, circumferential neck thrombus >50%	Talent, Zenith	77.5	95.9	41.9
Jim J [31]	2010	United States of America	2002–2003	53	60	wide neck: diameter >28 mm; angulated neck: >60 degrees; short neck: length <15 mm; significant thrombus: >50% of neck circumference; reverse tapered neck: neck dilated >2 mm within 10 mm; neck bulge: focal neck enlargement >3 mm within 15 mm	Talent	76.5	96.2	0
Troisi N [32]	2010	Germany	2007–2009	106	9	short neck: <10 mm; neck bulge: focal dilatation >3 mm within 15 mm; tapered neck: enlargement >2 mm within 10 mm; angulated neck: >60 degrees; neck thrombus: >50% of neck circumference	Endurant	73.6	100	9
Aburahma AF [33]	2011	USA	2004–2010	149	22	short neck: <10 mm; angulated neck: >60 degrees; wide diameter: >28 mm; calcified neck: >50% of neck circumference; neck thrombus: >2 mm thick; reverse tapered neck: neck dilatation >2 mm within the first 10 mm	AneuRX, Excluder, Zenith, Talent	74.3	99	22
Georgiadis GS [34]	2011	Greece	2009–2010	34	12.44	neck length between 5 and 12 mm; neck angulation between 60 and 90 degrees	Endurant	72.8	100	8.8
Hoshina K [35]	2011	Japan	2006–2008	49	26	short neck: <15 mm; angulated neck: >60 degrees	Zenith, Excluder	NA	NA	51
Hyhlik-Dürr A [36]	2011	Germany	2008–2009	50	15	short neck: <15 mm	Endurant	75	96	4
Rouwet EV [37]	2011	International	2007–2008	80	12	infrarenal angulation >60 degrees	Endurant	76	100	0
Torsello G [38]	2011	Germany	2007–2010	56	12	short neck: <10 mm; angulated neck: >60 degrees	Endurant	75.3	100	1.8
Van Keulen JW [39]	2011	The Netherlands	2007–2009	19	12	any deviation from single-device IFU	Endurant	73	100	0
Lee M [40]	2012	Republic of Korea	2007–2010	19	18.7	angulated neck: >60 degrees; conical neck: diameter at 15 mm below the lowest renal artery >10% larger than the diameter at the lowest renal artery	Zenith, Talent	73.3	100	NA
Hager ES [41]	2012	United States of America	2002–2009	84	18.5	short neck: <15 mm	Excluder, Zenith	75.5	100	16.6
Kvinlaug KE [42]	2012	Canada	2008–2010	37	6	short neck: <15 mm; wide neck: >28 mm; angulated neck: >60 degrees	Endurant	75.3	100	NA
Setacci F [43]	2012	Italy	2010	72	1	hourglass neck; angulated neck: >60 degrees; short neck: <15 mm; thrombosed neck: >50% of the neck circumference; reverse conical neck: dilatation > 2 mm within 10 mm; barrel neck: focal enlargement >3 mm within 15 mm	Endurant	77	100	11.11
Stather PW [44]	2012	United Kingdom	1999–2010	199	48	angulated neck: >60 degrees; short neck: <15 mm; wide neck: >28 mm; thrombosed neck; flared neck	Zenith, Talent, Excluder, Endurant, Jotec Tube	73.9	98	NA
Antoniou GA [45]	2013	International	NA	60	18	short neck: <15 mm; angulated neck: >60 degrees	Endurant, Zenith	74	100	8
Mwipatayi BP [46]	2013	Australia	2008–2011	31	20	short neck: <10 mm; angulated neck: >60 degrees; reverse tapered neck: diameter >2 mm for every 5 mm distal from the most caudal renal artery	Endurant	75	100	12.9
Shintan T [47]	2013	Japan	2007–2011	20	25.7	short neck: <10 mm; angulated neck: >60 degrees; reverse tapered neck: dilation >2 mm within the first 10 mm; thrombosed neck: thrombus in the first 10 mm of the neck, with thickness >2 mm and covering >25% of the circumference	Excluder, Zenith	75.6	100	10
Ierardi AM [48]	2014	Italy	2009–2011	36	27.7	short neck: between 7 and 10 mm	Ovation	73.6	100	0
Igari K [4]	2014	Japan	2008–2010	12	25	short neck: <15 mm; angulated neck: >60 degrees	Excluder, Zenith, Powerlink	77.5	NA	0
Iwakoshi S [49]	2014	Japan	2009–2011	44	120	short neck: <15 mm; angulated neck: >60 degrees; reverse tapered neck: dilation >2 mm within the first 10 mm	Zenith	77	92.1	16
Setacci F [50]	2014	Italy	2010	72	24	hourglass neck; angulated neck: >60 degrees; short neck: <15 mm; thrombosed neck: >50% of the neck circumference; reverse conical neck: dilatation > 2 mm within 10 mm; barrel neck: focal enlargement >3 mm within 15 mm	Endurant	77	NA	NA
Speziale F [7]	2014	Italy	2010–2011	133	24	noncylindrical neck: hourglass, reverse conical neck (dilation > 2 mm within 10 mm), or barrel neck (focal enlargement >3 mm within 15 mm); angulated neck: >65 degrees; short neck <15 mm, wide neck: >28 mm	Endurant, Excluder, Zenith	NA	100	12
Kaladji A [51]	2015	France	1998–2012	170	38	wide neck: need for a stent graft >32 mm in diameter	Talent, Zenith, Excluder, Anaconda, Endurant, Vanguard, AFX, AneuRx, Zenith, Lifepath	75	100	0
Saha P [52]	2015	United Kingdom	2006–2008	27	72	wide neck: need for a stent graft >36 mm in diameter	Zenith	76	93	0
Cerini P [53]	2016	Italy	2005–2013	90	37	any deviation from single-device IFU	Zenith, Endurant, Evita	75.8	95.3 *	1.1
de Donato G [54]	2016	Italy	2010–2012	161	32	short neck: >7 mm; thrombosed neck: >50% of the neck circumference; calcified neck: >50% of the neck circumference	Ovation	75.2	99.3	0.6
Gallitto E [55]	2016	Italy	2005–2010	60	51.4	short neck: <10 mm	Zenith, Endurant	74.9	95 *	7
Gimenez-Gaibar A [5]	2016	Spain	2006–2013	52	24	short neck: 15 mm; angulated neck: >60 degrees thrombosed neck: >50% of the neck circumference; calcified neck: >50% of the neck circumference	Excluder, Talent, Anaconda, Zenith, Endurant	75.9	100	13.4
Sirignano P [56]	2016	Italy	2012–2014	21	9	noncylindrical neck: hourglass, reverse conical (dilated >2 mm within 10 mm, barrel (focal enlargement >3 mm within 15 mm); angulated neck: >65 degrees; short neck: <10 mm; enlarged neck: diameter >30 mm; thrombosed neck: mural thrombosis >3 mm	Ovation	75.6	100	0
de Donato G [57]	2017	Italy	2010–2012	89	32	short neck: <7 mm	Ovation	76.4	97.7	2.2
Gargiulo M [58]	2017	International	2009–2012	118	37.9	wide neck: >28 mm	Zenith, Endurant, Excluder, Ovation, Anaconda	73.9	98	5
Kontopodis N [59]	2017	Greece	NA	106	18	short neck: >7 mm	Ovation	NA	97.2	0
Lee JH [60]	2017	Republic of Korea	2010–2013	38	1	conical neck: neck coefficient calculated using the following formula (diameter, D): Arctangent ([D3-D1]/[neck length]) × 180/π, if the absolute value of the neck coefficient was >10, it was defined as conical or inverted conical	Zenith, Endurant	73.8	100	23.7
Pitoulias GA [61]	2017	International	2007–2015	156	41.1	short neck: <15 mm; angulated neck: >60 degrees; wide neck: <32 mm; circumferential thrombus with >2-mm thickness; circumferential calcification >50%; reverse tapered neck: neck dilation >2 mm within 10 mm; neck bulge	Endurant	73.4	100	2.5
Sirignano P [62]	2017	Italy	2012–2015	156	20.4	short neck: <10 mm; noncylindrical aortic neck	Ovation	74.83	100	10.25
Reyes Valdiva A [63]	2017	International	2007–2015	73	30	need for a stent graft >36 mm in diameter; any deviation from single-device IFU	Endurant	74.4	98.6	6.8
Aburahma AF [10]	2018	United States of America	2003–2015	33	31.8	wide neck: >31 mm	Excluder, Zenith, AneuRX	74.7	100	NA
Bryce Y [64]	2018	United States of America	2004–2013	125	47.3	short neck: <10 mm; angulated neck: >60 degrees; reverse conical neck (neck dilated > 2 mm within 10 mm, barrel neck (focal enlargement > 3 mm within 15 mm); thrombosed neck: >50% of the neck circumference; calcified neck: >50% of the neck circumference	Endurant, Excluder, Zenith, Ovation, AFX	75.4	100	20
Greaves NS [65]	2018	United Kingdom	2012–2017	52	21.5	short neck: between 7 and 10 mm	Ovation	75.7	100	1.9
Howard DPJ [66]	2018	International	2011–2017	1189	60	wide neck: >25 mm	Excluder	73.9	99.9	10.4
Oliveira NFG [15]	2018	International	2009–2011	97	48	wide neck: >30 mm	Endurant	73.3	100	NA
Zhou M [67]	2018	China	2010–2015	323	36	short neck: <15 mm; very short neck: <10 mm; wide neck: >28 mm; conical neck: neck dilated over 2 mm within 10 mm below; angulated neck: >60 degrees thrombosed neck: the widest part of thrombus (≥2 mm thick) covering at least 50% of the circumference; calcified neck: calcification accounting for more than or equal to 50% of proximal neck	Endurant, Excluder, Zenith	73	89.2	10.2
Kouvelos GN [68]	2019	Greece	2009–2016	64	24	wide neck: 29–32 mm	Endurant	72.7	100	1.5
McFarland G [69]	2019	United States of America	2000–2016	108	34.1	wide neck: >28 mm	Excluder, Zenith, Talent, Endurant, Ovation, AFX	76.5	NA	NA
Sirignano P [70]	2021	International	2017–2018	122	1	Anatomy outside IFU for any commercially available endografts, while inside the IFU for the Ovation stent graft	Ovation	78.65	100	0

* Failure due to unintentional renal artery coverage and occlusion.

**Table 2 jcm-11-04460-t002:** Thirty-day, and mean follow-up complications rates of the 52 studies selected for the review.

First Author	30 Days					Mean Follow-Up				
	Conversion to Open Repair (%)	Reintervention (%)	Migration (%)	Type Ia Endoleak (%)	AAA-Related Mortality (%)	Conversion to Open Repair (%)	Reintervention (%)	Migration (%)	Type Ia Endoleak (%)	AAA-Related Mortality (%)
Dillavou ED [24]	0	1.09	NA	2.18	1.09	0	8.8	0	2.18	1.09
Fairman RM [25]	NA	NA	NA	NA	NA	3.1	NA	13	10.5	NA
Choke E [26]	0	3	0	3	0	0	1.5	0	3	0
Cox DE [27]	NA	0	NA	0	NA	NA	10.5	5.26	5.26	NA
Mc Donnell CO [28]	0	2.17	0	2.17	0	NA	0	2.17	0	0
Abbruzzese TA [29]	NA	NA	NA	NA	NA	1.4	24	1.4	0.9	11
Chisci E [30]	0	0	0	4.1	0	2.7	20.3	2.7	5.4	4.1
Jim J [31]	0	0	0	0	0	0	0	2.7	2.7	2.7
Troisi N [32]	NA	1.3	NA	1.3	1.3	0	0	0	0.65	0.65
Aburahma AF [33]	0	1	0	1	0	0	7	1.3	11	1
Georgiadis GS [34]	NA	NA	NA	NA	NA	0	0	0	0	0
Hoshina K [35]	NA	NA	NA	NA	NA	NA	NA	NA	NA	0
Hyhlik-Dürr A [36]	0	2	0	6	0	0	0	0	0	0
Rouwet EV [37]	0	0	0	0	1.25	0	0	0	0	1.25
Torsello G [38]	0	1.8	0	3.6	0	0	1.8	0	3.6	1.8
Van Keulen JW [39]	0	0	0	0	0	NA	0	NA	0	0
Lee M [40]	NA	NA	NA	NA	NA	0	10.5	0	10.5	0
Hager ES [41]	0	1.2	0	7.14	0	0	0	0	2.4	0
Kvinlaug KE [42]	0	0	0	0	0	0	0	0	0	0
Setacci F [43]	0	0	0	0	0	NA	NA	NA	NA	NA
Stather PW [44]	NA	5	NA	2.5	0.5	NA	2.5	3	9.5	2
Antoniou GA [45]	NA	NA	NA	NA	NA	0	0	0	1.7	0
Mwipatayi BP [46]	0	0	0	0	0	0	0	0	0	0
Shintan T [47]	NA	NA	NA	NA	NA	0	0	25	0	0
Ierardi AM [48]	0	0	0	0	0	0	0	0	0	0
Igari K [4]	0	0	0	8	0	0	0	0	0	0
Iwakoshi S [49]	0	0	0	0	0	0	3.14	0	3.14	2.36
Setacci F [50]	NA	NA	NA	NA	NA	0	5.5	0	5.5	0
Speziale F [7]	0	0	0	0	0	0	7.7	4.6	3	0
Kaladji A [51]	0	8.3	0	4.1	0	0	24.1	0	13	3.5
Saha P [52]	0	0	0	3.7	0	0	7.4	0	7.4	7.4
Cerini P [53]	2.2	11.1	NA	8.8	0	0	0	0	0	0
de Donato G [54]	0	0	0	0.6	0	0	1.8	0	1.8	0
Gallitto E [55]	NA	1.5	NA	3	NA	NA	3	NA	1.5	3
Gimenez-Gaibar A [5]	0	1.9	0	1.9	0	0	4.5	0	2.2	0
Sirignano P [56]	0	0	0	0	0	0	0	0	0	0
de Donato G [57]	0	0	0	0	0	0	2.2	0	2.2	0
Gargiulo M [58]	NA	NA	NA	NA	NA	6	7	3	12	3.4
Kontopodis N [59]	1.9	NA	NA	NA	NA	0	0	0	0	NA
Lee JH [60]	NA	NA	NA	NA	NA	NA	NA	NA	NA	NA
Pitoulias GA [61]	0	1.2	0	1.9	0	0	1.2	0	1.2	0
Sirignano P [62]	0	0.7	0	1.3	0	0	0	0	0	0
Reyes Valdiva A [63]	0	0	0	0	0	0	0	0	0	1.3
Aburahma AF [10]	0	15.2	0	27.3	0	0	17.2	0	13.8	0
Bryce Y [64]	0	1.6	0	1.6	0	0.8	1.6	0	1.6	0
Greaves NS [65]	0	0	0	0	0	0	0	0	0	0
Howard DPJ [66]	NA	NA	NA	NA	NA	0.2	3	0.1	0.3	0
Oliveira NFG [15]	NA	NA	NA	NA	NA	NA	3.1	NA	7.6	1
Zhou M [67]	NA	NA	NA	NA	NA	NA	5.6	NA	7.1	NA
Kouvelos GN [68]	0	0	0	1.5	0	7.2	10.14	2.9	4.3	1.5
McFarland G [69]	NA	NA	NA	NA	NA	1.85	11.1	14.8	14.8	0
Sirignano P [70]	0	1.6	0	1.6	0	NA	NA	NA	NA	NA

NA: not available.

## Data Availability

Data sharing not applicable.

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
