# Peer review of "Is Evar Feasible in Challenging Aortic Neck Anatomies? A Technical Review and Ethical Discussion"

_jcm, 2022, doi:10.3390/jcm11154460_

Round 1

Reviewer 1 Report

1. The authors concluded that EVAR appeared to be a safe and effective procedures even in patients with hostile neck anatomy. However, it is insufficient to conclude that only by showing the outcomes of hostile neck anatomy. The data and outcomes in patients with favorable neck anatomy should be reviewed and compared with those with hostile neck anatomy.

2. Definition of hostile neck anatomy differed depending on papers, therefore the definition of each studies should be added in Table 1.

3. What was the cause of AAA related death in thirty-day results? Rupture? or Procedure-related complications?

Author Response

Reviewer 1

  1. The authors concluded that EVAR appeared to be a safe and effective procedures even in patients with hostile neck anatomy. However, it is insufficient to conclude that only by showing the outcomes of hostile neck anatomy. The data and outcomes in patients with favorable neck anatomy should be reviewed and compared with those with hostile neck anatomy.

We want to thank the Reviewer for his/her excellent comment, he/she pointed out the attention of one of the most interesting and “hot” aspect: the differences existing in outcomes between standard and challenging (or hostile) neck anatomies. Although we completely agree with the necessity to perform a such of a kind of analysis, we should admit that not all the included studies reported EVAR results in standard fashion. Therefore, we are not able to review the paper as per Reviewer’s suggestion. This aspect is now clearly reported in study limitations section: “First of all, present study is a narrative review and not a systematic review, consequently the statistical power of here presented data should be carefully evaluated. Moreover, not all included studies reported results on standard EVAR performed in standard anatomies, therefore a proper comparison was not made between patients presenting with standard and hostile anatomies treated by the same operators, in the same centeres, during the same period”.

  1. Definition of hostile neck anatomy differed depending on papers, therefore the definition of each studies should be added in Table 1.

We want to thank the Reviewer for his/her valuable suggestions, a new column was added to Table 1 to better define hostile anatomy in each analyzed study

  1. What was the cause of AAA related death in thirty-day results? Rupture? or Procedure-related complications?

We want to thank the Reviewer for his/her valuable suggestions. Cause of death, even AAA-related death, was not clearly specified in all studies analyzed. However, according to Reviewer’s comment, we are clearly reported in the revised text that no AAA suffered an immediate post EVAR rupture.

Reviewer 2 Report

The authors performed a literature review on EVAR in hostile aortic necks with a particular attention on the ethical implications in the clinical practice. The following are my comments and questions:

Introduction

Patients with challenging neck have other therapeutic options (BEVAR, FEVAR, sm-EVAR) , besides off-label EVAR and OR. This aspect should be included in the introduction.

Methods

Please introduce a paragraph regarding the statistical analysis. In the results session (i.e Page 6) there are reports regarding mean and SD. How was the statistical analysis performed? Did you performed a weighted mean?

Tables

The Tables extend in several pages; please include a table header in every page

Table 1:

typo “2007-201”, complete the study period of the line with reference 38

Table 2:

A part of the header is

please report in the table header the units ( %,, months, etc.)

typo: Reference number von Aburahma and Oliveira

Please include legend for the abbreviation NA

Results

Consider providing a flow chart of the study screening and study selection

Page 6: cit.“Technical success rates ranged from 93% to 100%;” This is not according to Table 1, several studies have technical success rate<93% and  technical success rate of Reference 67 amount to 89.2%. Please explain.

Endograft migration rate was not reported, according with results reported by 31 studies (Table 2).” The meaning of this sentence is unclear, please rephrase

The results your statistical analysis ( mean ± SD) should be reported in a separate table

Discussion

The present discussion focus exclusively on the ethical aspects. Please also discuss the results and their significance for the clinical practice.

The perioperative benefit deriving from lower perioperative morbidity and mortality for off-label EVAR have been suggested to be lost in long term. This aspect should be discussed.

Please present the limits of your study at the end of the discussion, with particular regard to the possible confounders and non-systematic review process

General considerations for the overall manuscript:

The authors review and summarize the literature focusing on the issue of elective EVAR outside of IFU because of challenging neck. In the published literaure, the topic of challenging neck have been quite extensively discuss and recently sistematically reviewed (i.e.  Eur J Vasc Endovasc Surg 2022 May;63(5):696-706.  doi: 10.1016/j.ejvs.2021.12.042.). However, this manuscript adds considerations regarding the ethical aspects resulting in this sense interesting and relevant for the clinical practice. The tables need minor corrections. It is of important to set an accurate layout for the tables, which are in the present form not clear to interpret for the reader.

Author Response

Reviewer 2

The authors performed a literature review on EVAR in hostile aortic necks with a particular attention on the ethical implications in the clinical practice.

The following are my comments and questions:

Introduction

Patients with challenging neck have other therapeutic options (BEVAR, FEVAR, sm-EVAR), besides off-label EVAR and OR. This aspect should be included in the introduction.

We want to thank the Reviewer for his/her valuable suggestions. According to his/her comment, a new paragraph was added to the revised text to better analyze this point: “In order to overcome this problem, several different technical solutions, such as parallel grafts, fenestrated or branched devices, have been developed as an alternative to standard EVAR grafts in patients with a “challenging neck”. However, all those solutions present a relevant risk of reintervention due to branch-related complications, and higher cost compared to standard EVAR; lastly custom-made device are not suitable for urgent/emergent cases

Methods

Please introduce a paragraph regarding the statistical analysis. In the results session (i.e Page 6) there are reports regarding mean and SD. How was the statistical analysis performed? Did you performed a weighted mean?

We want to thank the Reviewer for his/her valuable suggestions. No weighted statistical analysis was performed, and we were able to analyze adverse event rates as percentage reported in included study and we also reported the mean % value. No SD was analyzed (one mistake was emended in the revised text). Revised text was entirely revised according to reviewer comment and a new table was added.

Tables

The Tables extend in several pages; please include a table header in every page Added according to Reviewer’s suggestion

Table 1:

typo “2007-201”, complete the study period of the line with reference 38 Completed according to Reviewer’s suggestion

Table 2:

A part of the header is

please report in the table header the units (%, months, etc.) Added according to Reviewer’s suggestion

typo: Reference number von Aburahma and Oliveira Added according to Reviewer’s suggestion

Please include legend for the abbreviation NA Added according to Reviewer’s suggestion

Results

Consider providing a flow chart of the study screening and study selection Added according to Reviewer’s suggestion

Page 6: cit. “Technical success rates ranged from 93% to 100%;” This is not according to Table 1, several studies have technical success rate<93% and  technical success rate of Reference 67 amount to 89.2%. Please explain.

We want to thank the Reviewer for his/her valuable suggestion, and we apologize for a mistake in reporting data. Paragraph was emended in the revised text.

“Endograft migration rate was not reported, according with results reported by 31 studies (Table 2).” The meaning of this sentence is unclear, please rephrase. We want to thank the Reviewer for his/her valuable suggestion. Sentence was rephrased accordingly.

The results your statistical analysis (mean ± SD) should be reported in a separate table. Added according to Reviewer’s suggestion; no SD values are reported as above mentioned

Discussion

The present discussion focus exclusively on the ethical aspects. Please also discuss the results and their significance for the clinical practice. The perioperative benefit deriving from lower perioperative morbidity and mortality for off-label EVAR have been suggested to be lost in long term. This aspect should be discussed.

We want to thank the Reviewer for his/her kind words and valuable suggestions. As the Reviewer pointed out, the immediate clinical benefit of standard EVAR even in challenging anatomies has been confirmed by Patel in his recent systematic review and network meta-analysis (EJVES 2022 May;63(5):696-706). Nevertheless, a new paragraph has been added in discussion to better express immediate advantages and late risks of performing related to standard EVAR performed in challenging anatomies: “Consistent with those findings, a recently published systematic review confirmed a significant reduction in perioperative mortality for EVAR compared to open repair without showing differences in AAA-related mortality at mid-term follow-up. However, risk of reinterventions still represents a major issue in those kinds of procedures and should be properly assessed in decision making process”.

Please present the limits of your study at the end of the discussion, with particular regard to the possible confounders and non-systematic review process

We want to thank the Reviewer for his/her valuable suggestion a Study limitations paragraph was incorporated into the revised text: “First of all, present study is a narrative review and not a systematic review, consequently the statistical power of here presented data should be carefully evaluated. Moreover, not all included studies reported results on standard EVAR performed in standard anatomies, therefore a proper comparison was not made between patients presenting with standard and hostile anatomies treated by the same operators, in the same centeres, during the same period”.

General considerations for the overall manuscript:

The authors review and summarize the literature focusing on the issue of elective EVAR outside of IFU because of challenging neck. In the published literaure, the topic of challenging neck have been quite extensively discuss and recently sistematically reviewed (i.e.  Eur J Vasc Endovasc Surg 2022 May;63(5):696-706.  doi: 10.1016/j.ejvs.2021.12.042.). However, this manuscript adds considerations regarding the ethical aspects resulting in this sense interesting and relevant for the clinical practice. The tables need minor corrections. It is of important to set an accurate layout for the tables, which are in the present form not clear to interpret for the reader.

We want to thank the Reviewer for his/her kind words and valuable suggestions

Round 2

Reviewer 1 Report

Thank you for revision according to reviewer's comments.